# Optimization of Carbon Nanotube-Coated Monolith by Direct Liquid Injection Chemical Vapor Deposition Based on Taguchi Method

**Omar Qistina** [1,2,3] ID, **Ali Salmiaton** [1,*] ID, **Thomas S.Y. Choong** [1], **Yun Hin Taufiq-Yap** [4] and **Shamsul Izhar** [1] ID

[1] Sustainable Process Engineering Research Group, Department of Chemical and Environmental Engineering, Faculty of Engineering, Universiti Putra Malaysia, UPM Serdang 43400, Selangor, Malaysia; qistina71@uitm.edu.my (O.Q.); csthomas@upm.edu.my (T.S.Y.C.); shamizhar@upm.edu.my (S.I.)

[2] Centre of Foundation Studies, Universiti Teknologi MARA, Cawangan Selangor, Kampus Dengkil, Dengkil 43800, Selangor, Malaysia

[3] Faculty of Chemical Engineering, Universiti Teknologi MARA, Shah Alam 40450, Selangor, Malaysia

[4] Catalysis Science and Technology Research Centre, Faculty of Science, Universiti Putra Malaysia, UPM Serdang 43400, Selangor, Malaysia; taufiq@upm.edu.my

\* Correspondence: mie@upm.edu.my; Tel.: +60-397-696-297

**Abstract:** Carbon nanotubes (CNTs) have the potential to act as a catalyst support in many sciences and engineering fields due to their outstanding properties. The CNT-coated monolith was synthesized over a highly active Ni catalyst using direct liquid injection chemical vapor deposition (CVD). The aim was to study the optimum condition for synthesizing CNT-coated monoliths. The Taguchi method with $L_9$ ($3^4$) orthogonal array design was employed to optimize the experimental conditions of CNT-coated monoliths. The design response was the percentage of carbon yield expressed by the signal-to-noise (S/N) value. The parameters including the mass ratio of Ni to citric acid (Ni:CA) (A), the injection rate of carbon source (B), time of reaction (C), and operating temperature (D) were selected at three levels. The results showed that the optimum conditions for CNT-coated monolith were established at $A_1B_2C_1D_2$ and the most influential parameter was D followed by B, C, and A. The ANOVA analysis showed the design was significant with R-squared and standard deviation of the factorial model equal to 0.9982 and 0.22, respectively. A confirmation test was conducted to confirm the optimum condition with the actual values of the average percentage of carbon yield deviated 1.4% from the predicted ones. The CNT-coated monoliths were characterized by various techniques such as field emission scanning electron microscopy (FESEM), energy dispersive X-ray spectroscopy (EDX), X-ray diffraction (XRD), and Raman spectroscopy.

**Keywords:** carbon nanotubes; carbon vapor deposition; optimization and Taguchi method

## 1. Introduction

Carbon plays a dual role as a catalyst or catalyst support for many chemical reactions due to its outstanding properties such as large surface area, high porosity, excellent electron conductivity, and relative chemical inertness [1]. Graphene and carbon nanotubes (CNTs) are the most investigated carbon materials which build broad interest in science and engineering. Both these carbon materials present remarkable physical, mechanical, thermal, and optical properties. The graphene is the basic structure of graphite, fullerenes, and CNTs with a planar sheet of $sp^2$-bonded carbon atoms densely packed in a honeycomb crystal lattice [2]. The fullerenes which are otherwise called bucky balls are

essentially closed hollow cages made up of carbon atoms. Moreover, CNTs are cylindrical in structure consisting of a hexagonal arrangement of hybridized carbon atoms.

The CNTs have raised much interest especially due to their outstanding properties such as high surface area, tubular structure, high strength, stiffness, and unique electronic properties [3–5]. The first exploration of CNTs was by Iijima in 1991 who introduced the fascinating forms of CNTs as the multi-walled carbon nanotubes (MWCNTs) [6] and, subsequently, the single-walled nanotubes (SWCNTs) [7,8]. These unique CNT structures have been discovered in several fields such as in medical sciences [9,10], electronics [11], optics [12], and composite materials [13,14]. The obtaining CNTs have promoted high absorption property, controllable mesoporosity, specific metal support interactions, yielding improved catalytic activity, and selectivity [15].

Typically, amongst the CNTs' synthesis method, chemical vapor deposition (CVD) is one of the methods applicable for industry owing to its ease and economy of production at a larger scale. Furthermore, this synthesis method is suitable in terms of product quality and quantity. The CVD process can also be modified to employ an energy source such as microwave, inductively coupled plasma CVD, low pressure, hot filament, alcohol catalytic with direct gas or liquid injection CVD, and others as reported in initiating the CNT growth. Moreover, there are various important parameters in synthesizing CNTs which are metal catalyst, metal support, promoters, and reaction conditions. These parameters playing a crucial role in the growth of different morphology of the CNTs. Therefore, the properties and end use applications of CNTs are tobe significant based on distinguished CNTs structure and chirality. Moreover, the reaction conditions are one of the crucial parameters in deciding the types of CNT formation and its yield including reaction temperature, reaction time, and flowrate, or ratio of carbon source and hydrogen gas [5].

As one of important parameters, the presence of catalyst may determine the mode of CNT growth on the support. Previous studies have shown that almost all metals from Group VIII transition metals namely Ni, Fe, and Co are active toward the catalytic process compared to noble metal series. Amongst these metals, Ni catalysts extensively exhibited higher activity and low cost [3,4,16]. Typically, the metal catalyst will combine with carbon to grow CNTs. The metal nanoparticles act as seeds that makes carbon deposited on them become more stable and accumulated enough for crystalline growth. The interaction between metal catalyst and support effected the mechanism of CNT growth whether through tip growth or root growth. Most of the catalysts are commonly prepared by impregnation [17,18], injection, or floating catalyst, co-precipitation method [19], and others.

The growth of CNTs has been investigated on sheets, fibers, pellets, and films [20]. Some studies used monolith structure to improve the surface area and porosity. Therefore, the application of cordierite monolith as support could be an alternative option to enhance the growth of CNTs and optimize the use of the active phase by dispersing them on the monolith surface with a high surface area. The catalyst on a carbon monolith could be an alternative option to powder because they provide lower pressure drop, smaller diffusion resistance, and better mass and heat transfer [21].

The typical preparation of monolith basically focused on ceramic and metallic structures which is covered with a secondary support (washcoat). In most applications, the coating involved metal oxide, zeolites, and carbon supported onto the monolith [22]. In general, the carbon-based monoliths are considered to be carbon supported on or extruded into the monolith structure. Recently, the carbon-coated monolith has attracted great attention to become an alternative support material for example in catalytic reactions. The common techniques used to coat carbon onto a monolith were dip-coating into a liquid polymer [23], slurry coating [24,25], and carbon grown by heating carbon source with hydrogen in a quartz reactor [15]. The dip-coating into liquid polymer has a tendency to accumulate in the corners of the monolith channel, while the disadvantage of slurry coating is that it makes the powder agglomerate and become difficult to filter. Therefore, growing carbon in a quartz reactor using carbon sources and hydrogen gas could be a better coating technique.

The general objective of the study was focused on the synthesizing of a CNT monolith using liquid carbon sources injected through CVD system which catalyzed over nickel-coated monolith

structure. Therefore, this paper reports the optimization study of synthesizing of a CNT monolith via direct liquid injection CVD system designed by the Taguchi method. The percentage of carbon yield of each run was evaluated via ANOVA analysis and characterized using field emission scanning electron microscopy (FESEM), energy dispersive X-ray spectroscopy (EDX), X-ray diffraction (XRD), and Raman spectroscopy.

## 2. Results and Discussion

### 2.1. Analysis of the CNTs' Growth on Monolith

The experiments aimed to obtain a high yield of CNTs after the calcination process using direct liquid injection CVD based on Taguchi optimization method. Figure 1 shows the average percent yield and standard deviation of the CNTs' growth at various conditions as tabulated in Table 1. The significance of the top five average percentage growth of the CNTs is demonstrated by the experimental run of 2, 6, 5, 9, and 7, respectively. It clearly shows that these top five runs were conducted at 600 and 700 °C of operating temperature. The two highest average percentage growth were contributed by run 2 and 6 both of which react at 700 °C within 60 min and 30 min of reaction time, respectively. While, the third and fourth places require more than 60 min for CNT growth at 600 °C. Thus, the first interpretations from the data analysis in Table 1 explained that 700 °C of operating temperature could achieve a great CNT growth within 60 min. Furthermore, the other parameters such as mass ratio of Ni to CA and injection rate of carbon source showed the highest average CNT growth was 1:1 Ni to citric acid mass ratio and 5 mL/h injection rate. The standard error of all runs was in the range of 0.038–0.743.

Figure 2 presents the atomic composition of the obtained CNT monolith catalyst. R1 and R2 display a high composition of carbon among others. The high formation of CNTs on the monolith surface might be due to good dispersion of Ni on the surface. These similar findings were also observed in Marconi et al. as they found the influence of the catalytic activity depended on the nature and interaction with the Ni active phase [26]. The cordierite monolith consisting of silica, alumina, and MgO had high oxygen storage capacities which are attributed to high Ni dispersion and improvement in their catalytic performances [27]. Therefore, high amounts of carbon deposited onto monolith surface from the synthesis process of CNTs were successfully catalyzed by Ni metal. The mass ratio of Ni to CA of 1:1 exhibits the highest average percentage growth of the CNTs followed by 1:3 and 1:5 according to the five top significant ones as shown in Table 1. This showed that less citric acid acted as a better chelating agent because higher concentration could initiate corrosion of the monolith surface.

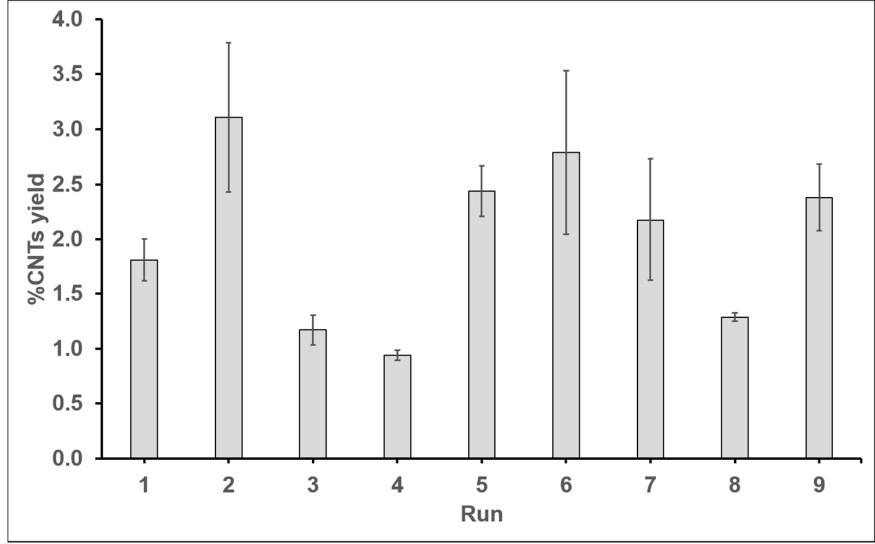

**Figure 1.** Carbon nanotube yield and standard error for each run of experiments.

**Table 1.** Experimentally measured values for carbon nanotube (CNT) growth and standard error of CNT monolith and signal-to-noise (S/N) ratio (Taguchi orthogonal array table of $L_9$ ($3^4$)).

| Parameters | | | | | * Carbon Yield (CNTs Growth) | | | | Response |
|---|---|---|---|---|---|---|---|---|---|
| Run | A | B | C | D | Yield 1 | Yield 2 | Average Yield | Standard error | ** S/N ratio |
| | | mL/h | min | °C | % | % | % | | dB |
| 1 | 1:1 | 1 | 30 | 600 | 2.005 | 1.617 | 1.811 | 0.194 | 5.158 |
| 2 | 1:1 | 5 | 60 | 700 | 3.785 | 2.433 | 3.109 | 0.676 | 9.852 |
| 3 | 1:1 | 10 | 90 | 800 | 1.304 | 1.036 | 1.170 | 0.134 | 1.360 |
| 4 | 1:3 | 1 | 60 | 800 | 0.986 | 0.894 | 0.940 | 0.046 | −0.538 |
| 5 | 1:3 | 5 | 90 | 600 | 2.670 | 2.211 | 2.440 | 0.229 | 7.749 |
| 6 | 1:3 | 10 | 30 | 700 | 2.046 | 3.533 | 2.789 | 0.743 | 8.910 |
| 7 | 1:5 | 1 | 90 | 700 | 1.620 | 2.732 | 2.176 | 0.556 | 6.753 |
| 8 | 1:5 | 5 | 30 | 800 | 1.325 | 1.248 | 1.287 | 0.038 | 2.189 |
| 9 | 1:5 | 10 | 60 | 600 | 2.079 | 2.684 | 2.381 | 0.303 | 7.536 |

A: mass ratio Ni:CA, B: injection rate of carbon source, C: reaction time, D: operating temperature. * The percentage of carbon yield calculated by Equation (3). ** The target signal-to-noise ratio is 'larger-is-better' (Equation (4)).

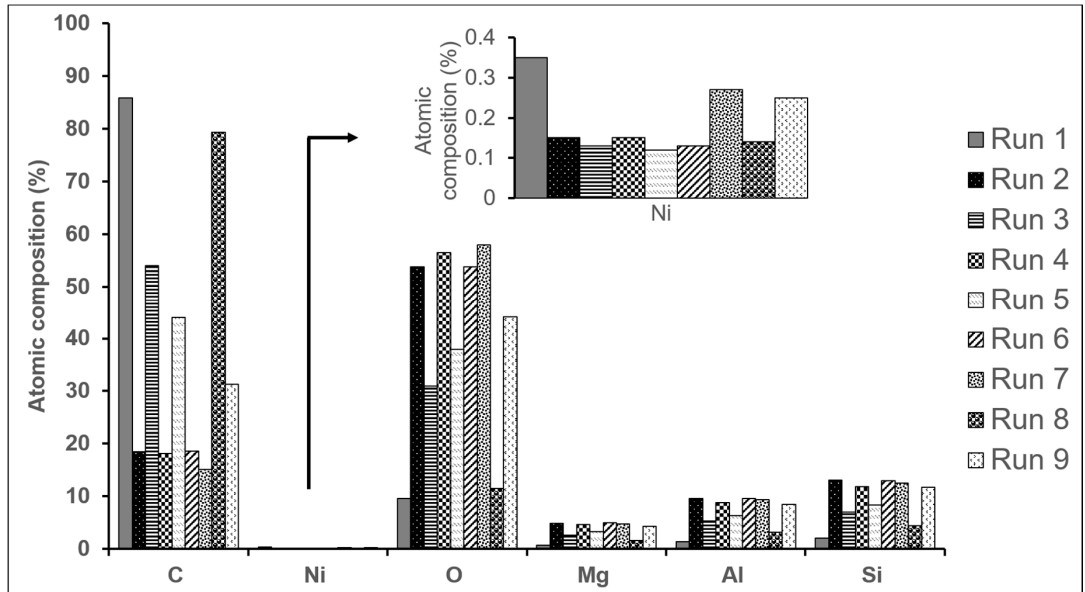

**Figure 2.** Energy-dispersive X-ray spectroscopy (EDX) of CNT monolith for each run of the Taguchi-based design.

As illustrated in Figure 3, the FESEM images show the CNTs deposited on the monolith surface. From the images, the yarn-like CNTs' growth can be observed. The CNTs' structure was very disordered and randomly oriented. As can be seen, R3, R4, and R8 show less growth of CNTs at 800 °C operating temperature even though they varied by other parameters. Likewise, Du and Pan investigated whether the growth of CNTs directly on nickel substrate has a strong effect of temperature on nucleation and CNT growth behavior with respect to reaction temperature and growth sites [28]. At lower temperatures the nickel nanoparticles act as nucleation sites for CNT growth. However, at high temperatures, the CNTs' growth only nucleates from grain boundaries and defective sites. Similar finding by Noda et al. in their study found that the number of CNTs decreased at high temperature for 1% and 5% concentrations of Ni due to sintering process over the different catalysts and supports the formation of SWCNTs and MWCNTs at different reaction temperatures (625–800 °C) [29]. Moreover, Hoyos-Palacio et al. investigated the influence of temperature on the metal catalyst performances on CNT growth [30]. All the runs produced almost mesoporous CNTs ranging between 2 and 50 nm due to the particle size distribution of the nanotubes as shown in the histogram analysis using Image J software.

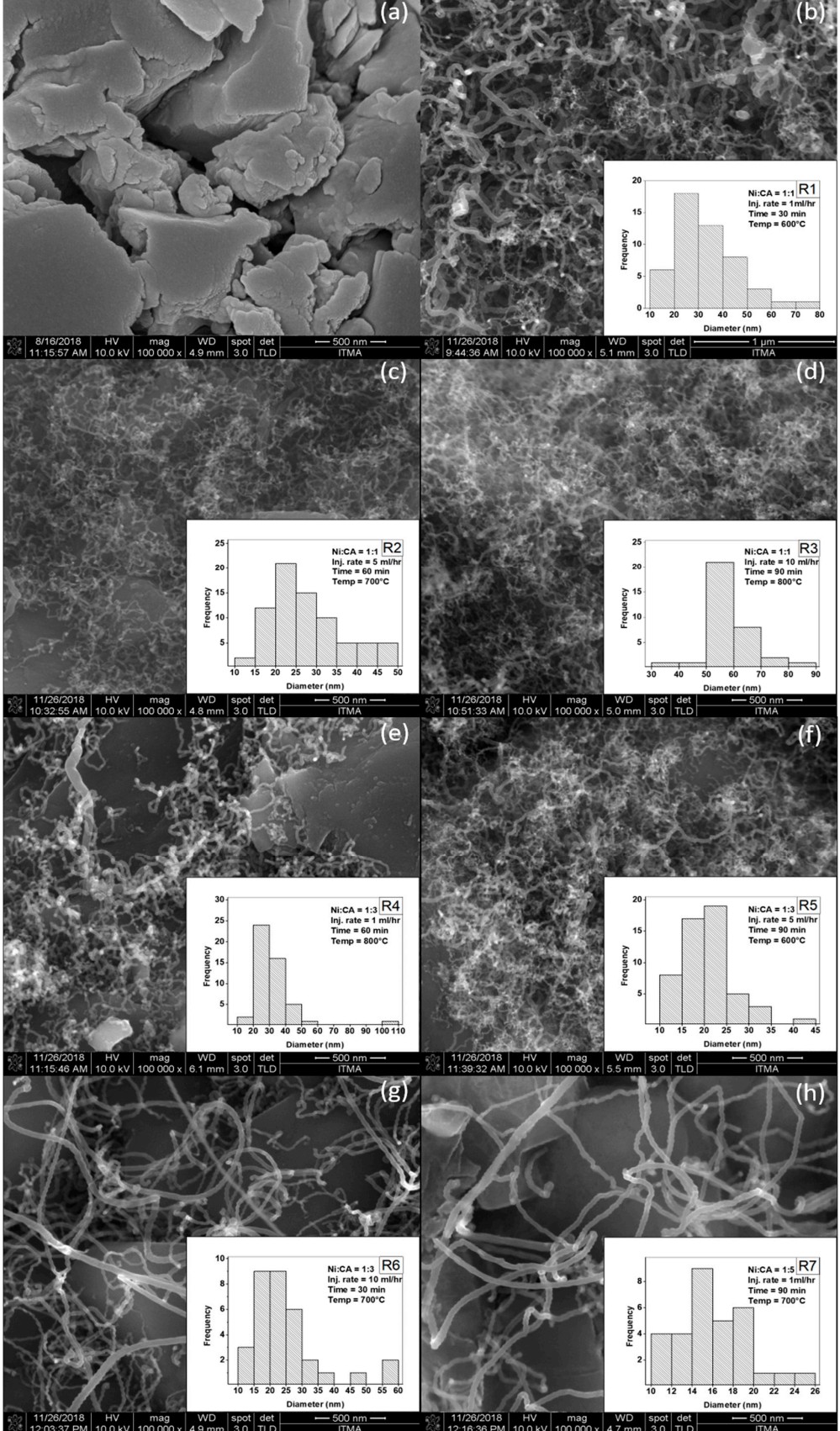

**Figure 3.** *Cont.*

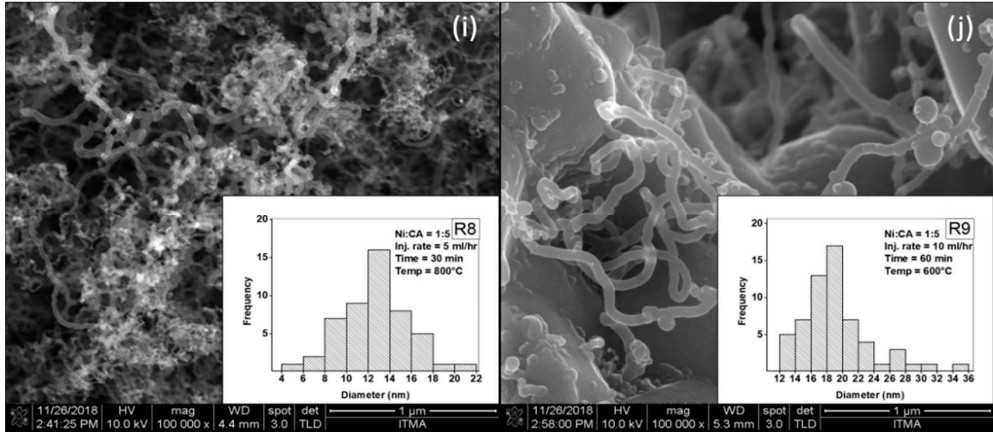

**Figure 3.** (**a**) Field emission scanning electron microscopy (FESEM) image of bare cordierite monolith; FESEM image and size distribution of CNTs monolith for (**b**) run 1; (**c**) run 2; (**d**) run 3; (**e**) run 4; (**f**) run 5; (**g**) run 6; (**h**) run 7; (**i**) run 8; and (**j**) run 9 of Taguchi-based design.

## 2.2. Response and Variance Analysis

The S/N ratios represent the log functions of desired output serve as an objective of function for optimization, assist in data analysis, and predict the optimum results. In this experiment, the Taguchi method applied the static problems to identify the optimum condition of the CNT growth process. The static problem could directly decide the best level of the control factor, which was the output target value (carbon yield). As shown in Table 2, the parameters for the synthesis of CNTs rank according to the delta which is the difference in the response of S/N ratio between the highest and lowest of S/N ratio within various levels of each parameter. Figure 4 is clearly describing the significant factors' effects in producing high carbon yields through the liquid injection CVD process. The optimum level for a factor is the level that contributes to the highest S/N ratio in the experiment. Therefore, the results from Table 2 and Figure 4 observed that the optimum condition of mass ratio of Ni:CA, injection rate of carbon, reaction time, and operating temperature are $A_1$, $B_2$, $C_1$, and $D_2$ which are expected to give the highest S/N ratio or the highest carbon yields.

**Table 2.** Response table of S/N ratios at different factor levels.

|       | Mass Ratio of Ni:CA | Injection Rate of Carbon | Reaction Time | Operating Temperature |
|-------|---------------------|--------------------------|---------------|-----------------------|
| Level | A | B | C | D |
| 1 | 5.457 | 3.791 | 5.419 | 6.814 |
| 2 | 5.374 | 6.597 | 5.617 | 8.505 |
| 3 | 5.493 | 5.935 | 5.287 | 1.004 |
| Delta | 0.119 | 2.806 | 0.329 | 7.501 |
| Rank | 4 | 2 | 3 | 1 |

ANOVA was used to determine the relative effect of the different factors via the decomposition of variance. The analysis of variance was obtained first by computing the sum of squares (SS) followed by adjusted mean squares (sum of squares divided with degree of freedom). All these values are tabulated in Table 3. Hence, the percentage contribution of each parameter can be calculated using Equation (1) [31]. Therefore, the larger the contribution of a factor to the total sum of squares, the larger the ability is of that factor to influence S/N ratio and become a more significant factor. Moreover, the larger the "F-value", the larger will be the factor effect in comparison to the error mean square or the error variance. Referring to the sum of squares in Table 3, the factor of D makes the largest contribution to the sum of the total sum of squares ($46.450/52.993 \times 100 = 87.65\%$). The factor of B

showed the second largest contribution (12.17%) to the total sum of squares, whereas the other two parameters A and C together made only less than 0.2%.

$$\text{Contribution (\%)} = \frac{SS_{adj}}{SS_{Tadj}} \times 100 \tag{1}$$

From Table 3, the selected factorial model of the design of experiment "F-value" of 565.79 implies the model is significant with "R-squared" 0.9982. There is only 0.01% chance that a "Model F-Value" this large could occur due to noise. Moreover, the values of "Prob > F" less than 0.0500 indicates that the model terms are significant. Meanwhile, the values greater than 0.1000 indicate the model terms are not significant. If there are many insignificant model terms, the model reduction may improve the design model. Therefore, from this design the injection rate of carbon source (B) and operating temperature (D) are significant model terms as both parameters serve the largest percentage contribution with the standard deviation as 0.22. The model shows a "Pred-R-Squared" of 0.9911 which is in reasonable agreement with the "Adj R-Squared" of 0.9965. Likewise, "Adeq Precision' as a measurement of the signal-to-noise ratio gives a ratio greater than four which is desirable. The ratio 63.957 indicates an adequate signal and proves this model can be used to navigate the design space.

The factorial model graph was plotted according to selected factors which were operating temperature and injection rate of carbon source as depicted in Figure 5. As can be seen, the vertical I-beam-shaped bars have the least significant difference (LSD) at a 95% confidence level (as default). This statistical plot further reinforced that the injection rate of carbon source (B) and operating temperature (D) influenced the CNT growth significantly, as proven from the ANOVA analysis. The advantage of the Taguchi orthogonal array design is that it gives the evaluation feature for examining aliases. In other words, it is good to proceed with a follow-up experiment to confirm and validate the effects of temperature and volume of carbon sources.

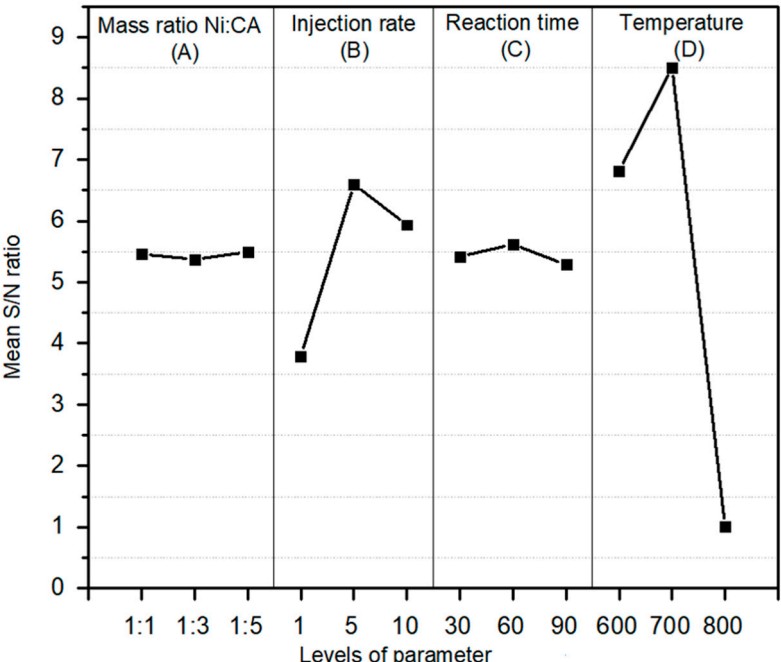

**Figure 4.** The mean S/N ratio of three levels at four different parameters assigned as mass ratio of Ni:CA (**A**); injection rate of carbon source (**B**); reaction time (**C**); and operating temperature (**D**).

**Table 3.** Summary of ANOVA for selected factorial model.

|  | Degree of Freedom (DF) | Sum of Squares (SS) | Adjusted Mean Square (Adj. SS) | Percentage Contribution (%) | F-Value | *p*-Value Prob > F |
|---|---|---|---|---|---|---|
| A | 2 | 0.022 | 0.011 | 0.02 |  |  |
| B | 2 | 12.910 | 6.450 | 12.17 | 138.060 | 0.0002 |
| C | 2 | 0.160 | 0.082 | 0.16 |  |  |
| D | 2 | 92.900 | 46.450 | 87.65 | 993.520 | <0.0001 |
| Model (B and D) | 4 | 105.810 | 26.450 |  | 565.790 | <0.0001 |
| Total (A, B, C, D) | 8 | 105.640 | 52.993 | 100.00 |  |  |
| R-Squared |  | 0.9982 (significant) |  |  |  |  |
| Adj. R-Squared |  | 0.9965 |  |  |  |  |
| Pred-R-Squared |  | 0.9911 |  |  |  |  |
| Adeq Precision |  | 63.957 |  |  |  |  |
| Std. Dev. |  | 0.22 |  |  |  |  |

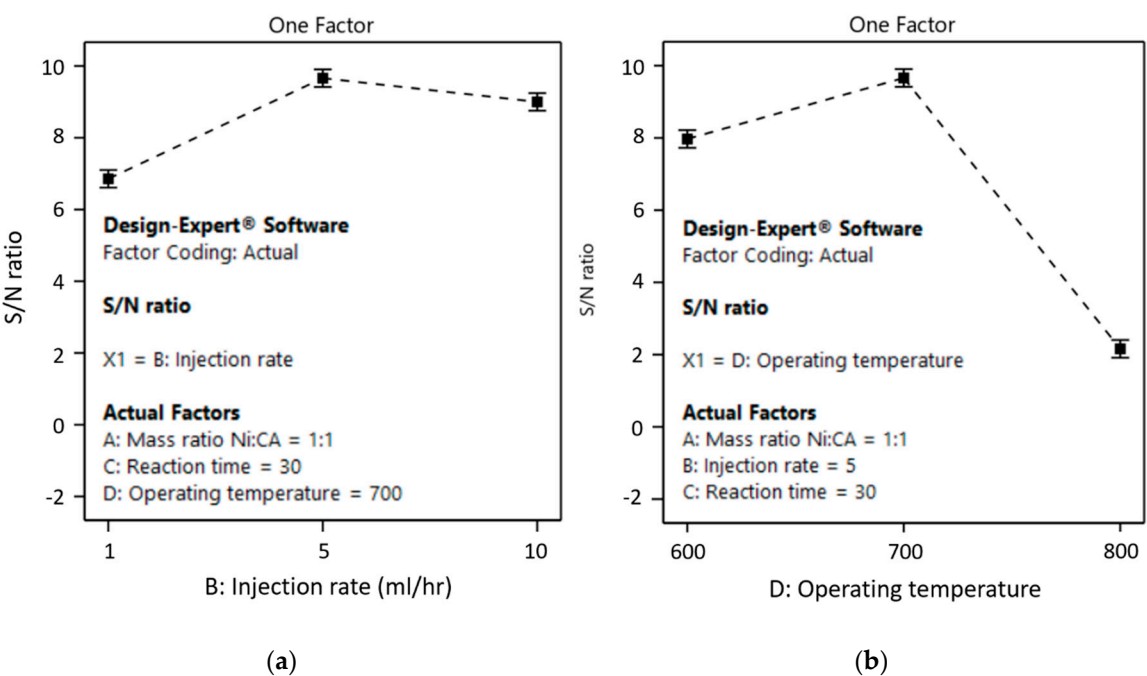

(**a**)          (**b**)

**Figure 5.** One plot factor of the main effect: (**a**) injection rate (**b**) operating temperature.

*2.3. Confirmation Test at Optimum Conditions*

Finally, based on the S/N ratio diagrams (Figure 5) with an optimum yield of CNT growth, a verification experiment was conducted according to the selected suggestions as tabulated in Table 4. The experiment was performed to investigate through a prediction of the performance of CNT growth yield at optimal process condition, and the confirmation experiment to verify the consistency of the optimization results via suggested condition by Taguchi method was carried out. The validation test was conducted with triplicates. Therefore, the acquired results exhibit a good agreement based on the actual S/N ratio of 9.5349 dB compared to the predicted result of the Taguchi method which was 9.6609 dB as the actual values of average percentage carbon yield deviated only 1.4% from the predicted ones.

**Table 4.** The optimal configuration with predicted and actual S/N ratio value.

| | | | | | Optimization Prediction | | Optimization Results | |
|---|---|---|---|---|---|---|---|---|
| | Parameter | | | | S/N Ratio Predicted | Yield CNTs Growth | S/N Ratio Actual | Yield CNTs Growth |
| Number | A | B | C | D | dB | % | dB | % |
| 1 | 1:1 | 5 | 30 | 700 | 9.6609 | 3.0415 | 9.7502 | 3.0726 |
| 1 | 1:1 | 5 | 30 | 700 | 9.6609 | 3.0415 | 9.3195 | 2.9240 |
| Average | | | | | 9.6609 | 3.0415 | 9.5349 | 2.9983 |
| Standard error | | | | | | | 0.1758 | 0.0607 |

### 2.4. Characterization of CNT Monolith at Optimum Conditions

The characterization of the CNT monolith at optimum condition evaluated by FESEM, size distribution, and EDX is depicted in Figure 6a–c, respectively. In determination of CNT diameter size distribution, 60 points have been selected and all points are displayed in Figure 6a. The FESEM micrograph showed the biggest size distribution of CNTs was about 32–34 nm in the range which is considered to be mesoporous nanotubes as shown in Figure 6b. Moreover, the CNT growth at optimum condition showed more uniform tubes by diameter due to fewer tube defects and the formation of less amorphous carbon as observed in the FESEM image. As shown in Figure 6c, the EDX spectrum presents 96.45% of carbon composition in the sample representing CNT growth on the surface of the monolith. The Ni composition disappeared due to high CNT growth which led to a possible covering of all Ni on the surface with the carbon.

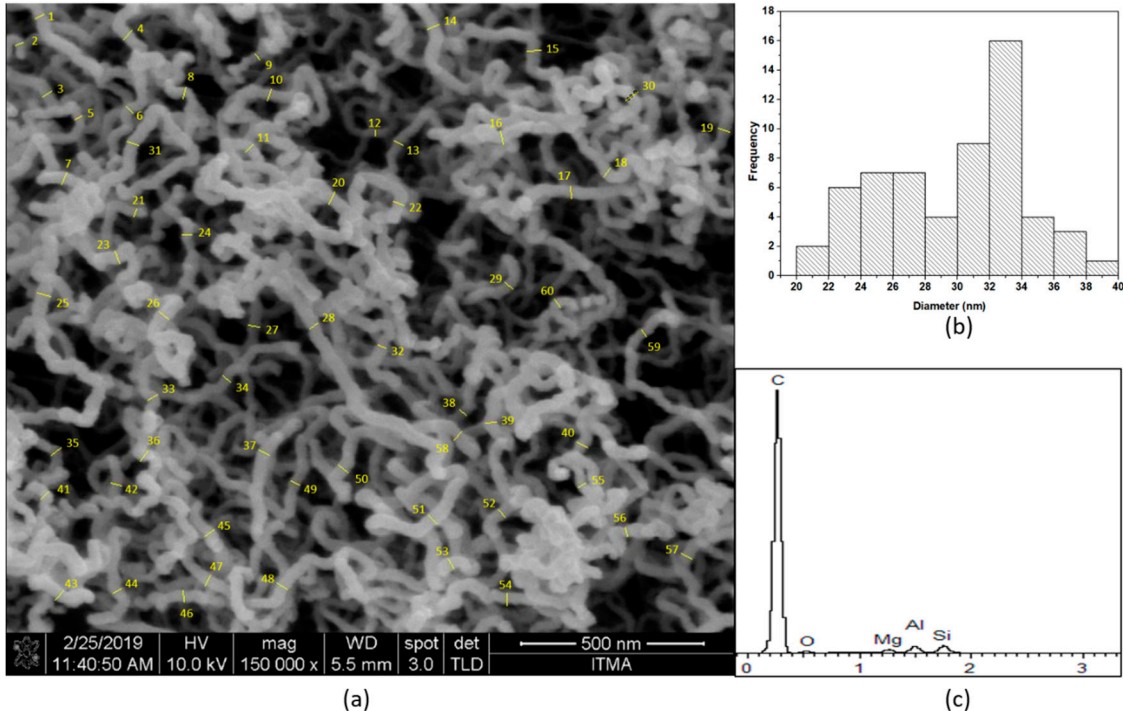

**Figure 6.** (**a**) Micrograph image of CNT monolith at optimum condition using FESEM; (**b**) size distribution of CNTs growth at optimum condition; and (**c**) EDX analysis.

Figure 7 represents the X-ray diffraction (XRD) patterns of cordierite monolithic support and the CNT growth at optimum conditions. The strong XRD patterns shown mostly correspond to cordierite phase (ICDD (International Centre for Diffraction Data) file No. 00-012-0303). The patterns show not much difference between the CNT phase patterns reflection at $2\theta = 26.46°(002)$ and $44.60°(004)$

according to ICDD file No.00-001-0646 with a hexagonal crystal system [32]. This could be because no acid-treated the cordierite monolith surface as citric acid is a weak acid. Therefore Al, Si, and Mg would not be leached when the material immersed in acidic condition. The Al and Mg would be leached easily in strong acid rather than Si as both have basic character [33,34]. Furthermore, a weak reflection at 2θ = 37.24°, 43.30°, and 62.70° indicate NiO (ICDD file No. 00-002-1216) while 2θ = 44.5° and 51.8° indicates Ni (ICDD file No. 00-004-0850). The crystallite size of the compound can be determined from the Scherer equation (Equation (2)) [35].

$$D = \frac{k\lambda}{\beta cos\theta},$$
(2)

where $\beta$ is the full width at half maximum (FWHM), $\theta$ is the diffraction angle, $\lambda$ is the X-ray wavelength (1.54178 Å), $D$ is the particle (crystallite) size, and $k$ is the Scherer constant (0.94).

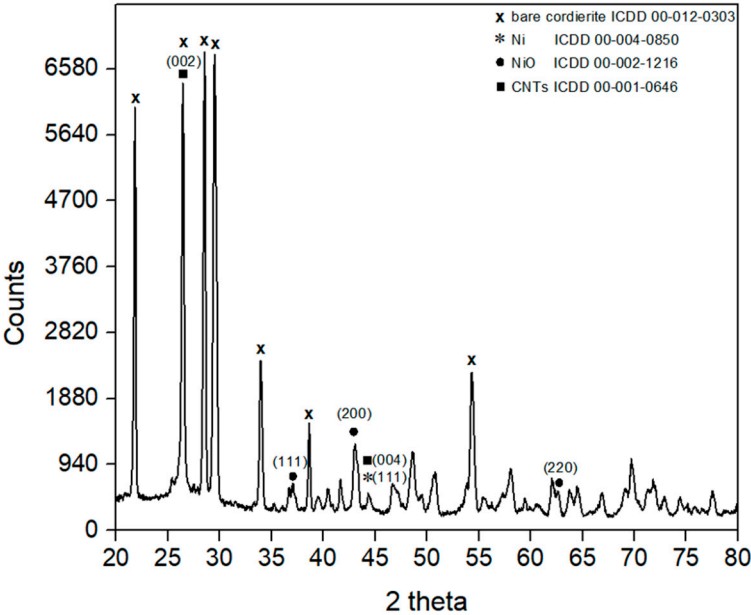

**Figure 7.** XRD pattern of CNT monolith at optimum condition.

Raman spectroscopy was used to characterize the carbonaceous materials in order to distinguish disordered and ordered crystal structures of carbon by illustrating the characteristic of the D and G bands of the samples. In this case, the D band represents the impurities and lattice disorderliness in the nanotubes. Meanwhile, the G band corresponds to degree of graphitization of CNTs such as purity and quality of CNTs resulting from the bond stretching of the $sp^2$ carbon pairs [36]. Moreover, the G' band indicates the graphene layer as single, double, or triple layer of the sample. Figure 8 shows the Raman spectra of the CNT growth onto monolith produced at optimum condition. The CNT monolith indicates higher G band about 7.84 in intensity compared to D band. The degree of disorder in the CNT monolith was 0.9921, expressed by the intensity ratio of D band to the G band ($I_D/I_G$) which is also an indicator of the $sp^3/sp^2$ carbon ratio. Therefore, the higher ratio indicates the presence of amorphous carbon and structural defects in the sample while the lower $I_D/I_G$ ratios show better graphitization of CNT structures.

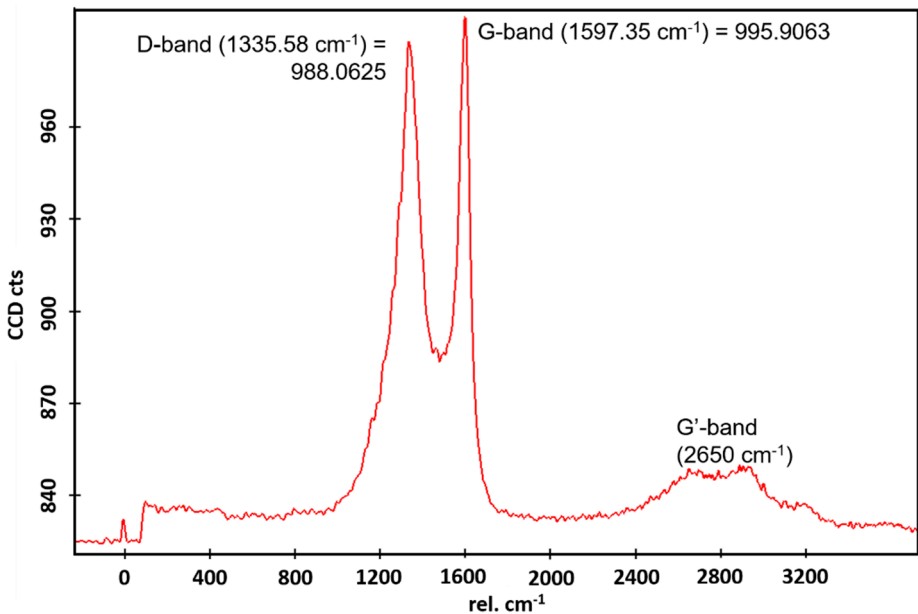

**Figure 8.** Raman spectra of CNT monolith.

## 3. Materials and Methods

### 3.1. Materials

The CNT-coated monolith (Figure 9b) was synthesized by direct liquid injection CVD method. The bare cordierite monolith as shown in Figure 9a was used as a support for the CNTs. The cordierite monolith with cell density of 400 cells per square inch (cpsi) consisting of a cylindrical ceramic-type monolith in the original length of 100 mm and diameter 25 mm was purchased from Beihai Huihuang Chemical Packing Co. Ltd., Beihai, Guangxi, China. The chemical compositions of the cordierite monolith were silica ($SiO_2$) 50.9% ± 1%; alumina ($Al_2O_3$) 35.2% ± 1%; magnesia (MgO) 13.9% ± 0.5%, and others <1%. As received, the monolith was cut into 10 mm length. Nickel nitrate hexahydrates (≥97%), furfuryl alcohol (≥98%), citric acid (≥99%), and ethanol (≥99%) obtained from Systerm, Classic Chemicals Sdn. Bhd., Malaysia were used as received without further purification. Ethanol solution was prepared by 50:50 volume ratio of ethanol and distilled water for monolith cleaning purpose. The nickel nitrate salts and citric acid were mixed in distilled water with various catalyst concentration solutions. The industrial nitrogen and purified hydrogen gas supplied by Alpha Gas Solution Sdn. Bhd., Malaysia were used as the carrier gases. Calcination in the air was conducted in a furnace, while reduction and calcination in carrier gases were conducted in a CVD horizontal quartz reactor as shown in Figure 10.

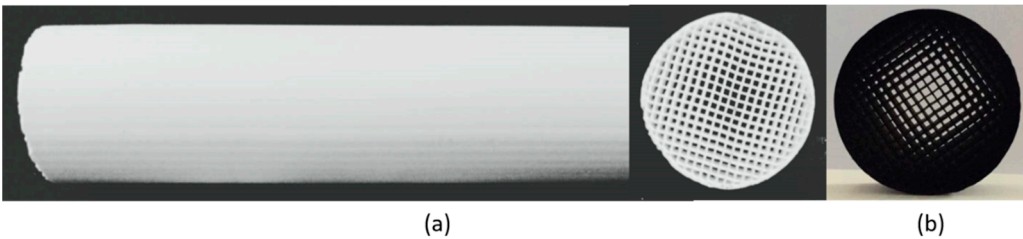

**Figure 9.** (**a**) Bare cordierite monolith and (**b**) CNT-coated monolith.

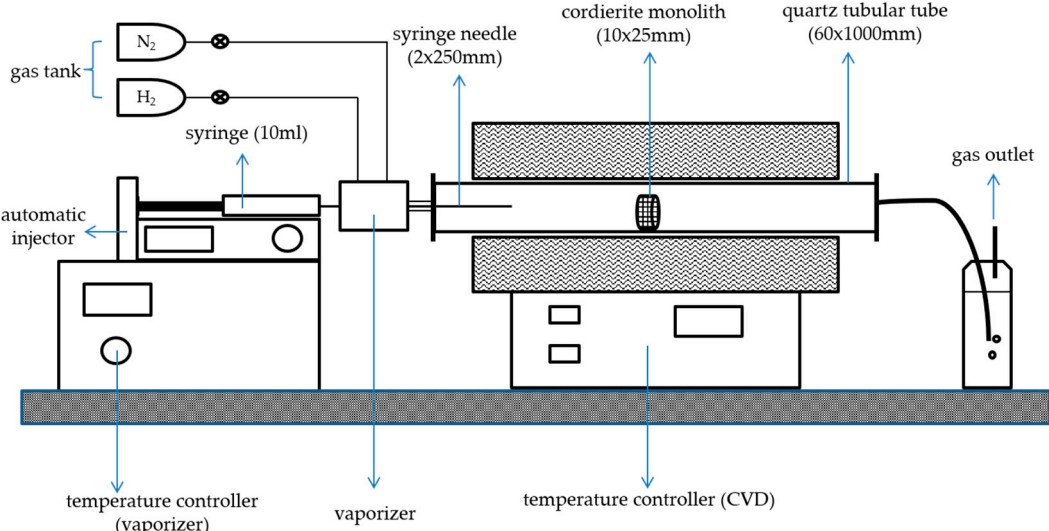

**Figure 10.** Schematic diagram of chemical vapor deposition (CVD) horizontal quartz reactor.

### 3.2. Preparation of the CNT-Coated Monolith

The experimental investigation was started with cleaning the cordierite monolith using ethanol solution to remove impurities in a sonicator for an hour at 80 °C. Then, the cleaned monolith was dried overnight in an oven at 110 °C. The mass ratio of Ni to citric acid was prepared by dissolving Ni nitrate salts and citric acid in distilled water. The metal catalyst, Ni was fixed at 1 wt.% of the mass of the clean monolith. Then, the monoliths were immersed in the metal/acid solution to disperse the metal on monolith surface and the citric acid was aimed to chelate more metal onto the surface. The immersion process was conducted for an hour and the remaining water was blown away using an air gun to remove the slurry excess before drying overnight at 110 °C in the horizontal position and was rotated continuously around its axis to prevent gravity from causing an uneven distribution. The blowing step was essential to obtain a homogeneous catalytic layer [21]. Furthermore, the immersed monolith was calcined at 450 °C in a furnace for three hours to convert $Ni(NO_3)_2$ into NiO. The immersion and calcination processes were repeated by two complete cycles to increase active phase dispersion and enhance CNT growth on the monolith support.

The CNT growth onto the monolith support was synthesized by employing the Taguchi method designed using Design of Experiment (DOE) as discussed in Section 3.3. The reaction parameters were selected for the design as listed in Table 5. Firstly, the NiO monolith was put into a tube of the CVD system and operating temperature and reaction time were set up as required by Taguchi method design as tabulated in Table 1. In the experiment, 50 standard cubic centimeters per minute (sccm) of $N_2$ gas flowed through the tube to remove oxygen inside the tube. Once the system reached the required temperature, the $H_2$ gas at 100 sccm slowly flowed into the reaction chamber to combine with $N_2$ gas. At this time, NiO on the monolith surface may convert into $Ni^{2+}$ acting as a catalyst in the growth of CNTs. An amount of furfuryl alcohol was injected at a few injection rates via a 20 inch-long syringe needle. The $H_2$ and $N_2$ gases were kept constant at 100 and 50 sccm respectively to make sure furfuryl alcohol droplets touch the monolith surface. After injection was completed, the $N_2$ gas flow speed was adjusted to 500 sccm and $H_2$ was 10% of the carrier gas. Then, the reaction time was started. When the reaction ended, the $H_2$ gas was shut off and $N_2$ was kept flowing at 50 sccm into the setup until it cooled down to room temperature.

**Table 5.** Parameters and their levels.

| | Parameters | Levels | | | References |
|---|---|---|---|---|---|
| A | Mass ratio of Ni to citric acid | 1:1 | 1:3 | 1:5 | [37] |
| B | Injection rate (mL/h) | 1 | 5 | 10 | [38] |
| C | Reaction time in CVD (min) | 30 | 60 | 90 | [39–41] |
| D | Operating temperature (°C) | 600 | 700 | 800 | [38,39,42–45] |

The flowchart of the experimental procedures of the CNT monolith is depicted in Figure 11. The particle size and the size of distribution of the CNT monolith were characterized using FESEM by Thermo Scientific NovaNano 230 at 100k magnification and atomic composition by energy-dispersive X-ray (EDX) by Oxford Instruments which was attached to the FESEM. Furthermore, the crystallinity of the substrate was analyzed by X-ray diffraction (XRD). The measurement was conducted by Philips Expert PW3040 XRD measurement operating at 40 kV and 40 mA with CuKα (λ = 1.5406) radiation source within the 2θ range from 20° to 80° with a 0.033°/min step size. The measurement for disordered carbon was evaluated using Raman spectroscopy by WITec Raman Microscope model Alpha 300R at 531.861 nm excitation wavelength within spectrum range about 100.36–3736.45.

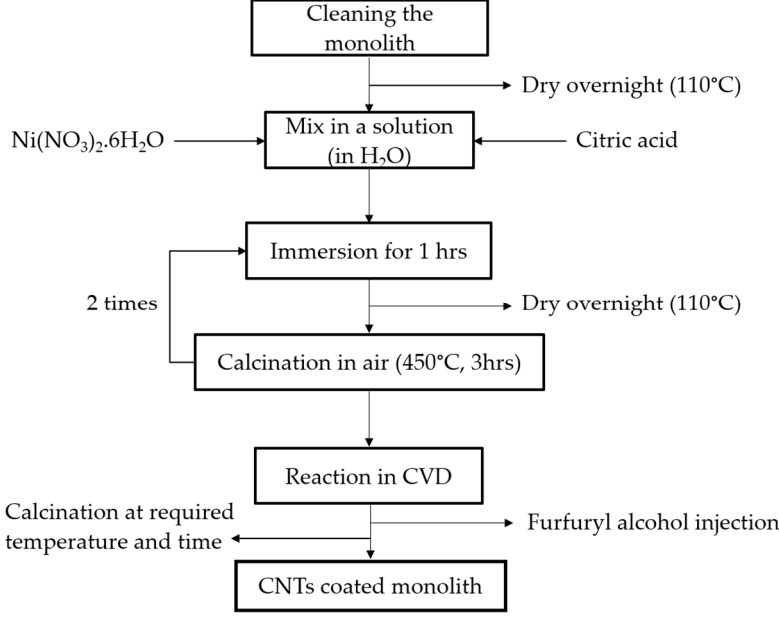

**Figure 11.** Flowchart for the synthesis of CNT monolith using a CVD system.

### 3.3. Taguchi Method of CNT-Coated Monolith

The Taguchi method was employed to provide an efficient way of designing a product that operates consistently and optimally over a variety of parameters. The development of the design could enhance on the experimental results and identify the controllable factors. This reduces variation of response results due to uncontrollable factors. The Taguchi method of CNT growth on monolith was conducted by the Design Expert version 7.0 from Stat-Ease Inc, Minneapolis, Minnesota, USA (2005). The orthogonal array table from the Taguchi method was used by choosing four parameters that could affect the CNT growth on the monolith. The orthogonal array of L₉ type was selected as presented in Table 1, in which L and subscript 9 mean the Latin square and the respective number of repetitions the experiment was carried out. The experiments were performed according to experimental procedures as presented in Figure 11 by adhering to the Taguchi orthogonal array table of $L_9$ ($3^4$) and were conducted in two replications for each determined condition resulting in a total of 18 runs. Both sets of

experiments were analyzed to determine the percentage of carbon yield on the monolith surface using Equation (3):

$$Carbon\ yield = \left[ \left( \frac{M_{total} - M_{cat}}{M_{cat}} \right) \times 100 \right], \tag{3}$$

where $M_{total}$ is the total mass of the final catalyst and carbon on monolith after the CVD reaction process and $M_{cat}$ is the initial mass of catalyst on monolith [46].

According to the Taguchi method, the yield of CNTs was selected as an output response which then employed a generic signal-to-noise (S/N) ratio as a quantitative measurement for determining the optimum conditions. The S/N proportion takes into consideration the influences of process parameters on the properties of the final product: CNTs yield. The changes in product properties minimized on noise factor (N-noise) and maximized on signal factors (S-signal). This S and N provides the statistical coefficient as S/N ratio which establish a logarithmic function of the desired output values. The main optimization target signal-to-noise ratio of this process utilizes a target which is 'larger-is-better' (Equation (4)) to get maximum yield of CNT growth. The S/N equations indicate the signal-to-noise ratio statistic where, n is the number of experiments and $y_i$ is the output response of the *i*-th experiment, $\bar{y}$ is average output response, and $\sigma$ is the standard deviation [47].

$$S/N_{larger-is-better} = -10 \log \left[ \frac{1}{n} \sum_{i=1}^{n} \frac{1}{y_i^2} \right] \tag{4}$$

In determining the significant parameters' effect on the production of high carbon yields, the response factor (S/N ratio) of each parameter was ranked according to delta value. The delta value can be calculated from the difference of the highest and the lowest S/N ratio value between level of each parameters in Taguchi method design as shown in Table 2.

## 4. Conclusions

In summary, the Taguchi method was employed to investigate the optimum parameters in synthesizing CNTs on monolith support. The mass ratio of Ni:CA (A), injection rate of carbon source (B), reaction time (C), and operation temperature (D) were used and arranged based on $L_9$ ($3^4$) orthogonal array of design of experiments at three levels. As a result, the identified optimum condition of the CNT growth on monolith should be performed at 1:1 mass ratio of Ni to CA ($A_1$), 5 mL/h injection rate of carbon ($B_2$), 30 min reaction time ($C_1$), and 700 °C operating temperature ($D_2$). This is because both the operating temperature of CVD and the injection rate of carbon source at level 2 showed the highest signal that should contribute to high CNT growth. While the other two factors namely reaction time and mass ratio of Ni and CA ratio should be performed at level 1 due to the absence of significant differences between other levels and minimum conditions should be more beneficial to the reaction. The ANOVA showed the design was significant, with R-Squared and standard deviation of the factorial models equal to 0.9982 and 0.22, respectively. Hence, the optimum conditions selected from the Taguchi analysis were also validated experimentally and the results showed a good agreement with the predicted conditions as the error of the carbon yield was less than 1.0%.

**Author Contributions:** A.S., T.S.Y.C., and Y.H.T.-Y. conceived the main idea and designed the study. O.Q. carried out the experiments and wrote the manuscript with input from A.S., T.S.Y.C., Y.H.T.-Y., and S.I. A.S. supervised the project and assisted in measurement and analysis. S.I. assisted in technical details for the experiment. All authors contributed in the overall project and input for writing the manuscript. All authors have read and agreed to the published version of the manuscript.

**Funding:** This work was financially supported by the Universiti Putra Malaysia, through grant no. GP-IPB/2016/9515202.

**Acknowledgments:** The authors would like to express their gratitude and acknowledgement to Muhammad Aidil Ibrahim, an enumerator at the Department of Chemical and Environmental Engineering, Faculty of Engineering for assisting in this research.

**Conflicts of Interest:** The authors declare no conflict of interest.

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
