# Peer review of "Optimization of Carbon Nanotube-Coated Monolith by Direct Liquid Injection Chemical Vapor Deposition Based on Taguchi Method"

_catalysts, doi:10.3390/catal10010067_

Round 1
Reviewer 1 Report
Dear Authors, I was interested in the manuscript's idea and would like to recommend the article to be published, but after strong improvements have to be performed.
General remarks
1) The figures 1, 2, 5, 6 are not looking properly, it's stretched. Please, make better images.
2) Figure 3 is very informative, but it is hard to see the numbers. It's necessary to change the design of the pictures.
3) Figure 4 does not give a chance to find a lot of additional information. The graphs look the same. Moreover, figure 8 contains a part of the data. I advise to remove the figure 4 or to point out the differences if any. Another option is to make the Supplementary materials section and the place there all additional data you are considering is necessary to publish.
4) There are too many details sometimes in the text. E.g. "... However, the CNTs growth at 600°C requires about 90 minutes and 60 minutes for the third and fourth highest average percentage growth..." That's why it is more difficult to show the main idea. The text should be improved.
Methods/data remarks
5) It is very important to write a description of SEM and powder XRD methods: what sort of instruments were used, typical settings, etc.
6) Figure 5 is nice, but it's always difficult to find (correctly) even linear trend via 3 points approximation only! There are two possible ways to make the results more accurate: a) to make experiments, a couple of additional points for the graphs b) to determine a deviation/standard error and to show it for the already existing points (to be sure the trends are fine).
7) The diffraction patterns have to be shown with the all strong lines identified (~22, 28, 29, 34, 38 of 2theta). Only a few medium intensity lines are allowed to be marked as "unknown phase" or "impurity". This remark is crucial. Intensity range/scale and units (e.g. 1000 a.u. or 10000, counts) should be inserted to the Y-axis, figure 8.
Hopefully, the remarks will help to make the publication with better quality, as well as it will help you to write new articles in the future.
Best wishes.
Author Response
Response to Reviewer 1 Comments
We thank the reviewer for all the comments.
Point 1: The figures 1, 2, 5, 6 are not looking properly, it's stretched. Please, make better images.
Response 1: The images have been improved.
See line 120-122, 137-138, 194-196, 232-234
Point 2: Figure 3 is very informative, but it is hard to see the numbers. It's necessary to change the design of the pictures.
Response 2: The design of the pictures has been changed and rearranged to get a better view. Moreover, we have added the bare monolith FESEM image to compare with the FESEM image for R1-R9.
See line 155-160
Point 3: Figure 4 does not give a chance to find a lot of additional information. The graphs look the same. Moreover, figure 8 contains a part of the data. I advise to remove the figure 4 or to point out the differences if any. Another option is to make the Supplementary materials section and the place there all additional data you are considering is necessary to publish.
Response 3: We have removed Figure 4 as advised by the reviewer.
Line 161-179
Point 4: There are too many details sometimes in the text. E.g. "... However, the CNTs growth at 600°C requires about 90 minutes and 60 minutes for the third and fourth highest average percentage growth..." That's why it is more difficult to show the main idea. The text should be improved.
Response 4: We have rewritten the sentences and improve the text to make it clearer.
Line 111-118
Point 5: It is very important to write a description of SEM and powder XRD methods: what sort of instruments were used, typical settings, etc.
Response 5: The description of FESEM and powder XRD methods have been reported in section 3.2 (Preparation of the CNTs coated monolith)
Line 354-360
Point 6: Figure 5 is nice, but it's always difficult to find (correctly) even linear trend via 3 points approximation only! There are two possible ways to make the results more accurate: a) to make experiments, a couple of additional points for the graphs b) to determine a deviation/standard error and to show it for the already existing points (to be sure the trends are fine).
Response 6: Figure 5 is related to Taguchi design based on L9(34) orthogonal array design which we selected only three levels (low, medium and high level) for each parameter. The figure depicted in Figure 5 used to describe the significant parameter effect in the Taguchi design based on calculation delta tabulated in Table 2. The standard deviation of the design has been reported in Table 3.
Line 399-402.
Point 7: The diffraction patterns have to be shown with the all strong lines identified (~22, 28, 29, 34, 38 of 2theta). Only a few medium intensity lines are allowed to be marked as "unknown phase" or "impurity". This remark is crucial. Intensity range/scale and units (e.g. 1000 a.u. or 10000, counts) should be inserted to the Y-axis, figure 8.
Response 7: The strong lines in the XRD patterns have been identified. Moreover, we also have done the correction to the Y-axis.
Line 265-284
Reviewer 2 Report
The manuscript written by Qistina Omar et.al. entitled ,,Optimization of carbon nanotubes coated monolith by direct liquid injection chemical vapor deposition based on Taguchi method’’ was read by me careful with lot of attention. The manuscript describes synthesis of CNTs by direct liquid injection chemical vapor deposition method. The work is interesting and fits well into a journal of Catalysts.
The work requires minor corrections:
The novelty of this work should be highlighted in the introduction part of the paper. image or images of the obtained material should be add to the work to present the quality of CNTs obtained during other conditions of the synthesis in order to compare and to show the differences. Raman results in relation to the purity of the obtained nanotubes would enrich the work. More discussion with the literature data can affect the improvement of scientific work - in particular as regards the characteristics of carbon nanotubes.Base on the above mentioned comments I would recommend the article for publication after minor revision.
Author Response
Response to Reviewer 2 Comments
We thank you for all the comments.
Point 1: The novelty of this work should be highlighted in the introduction part of the paper.
Response 1: We have highlighted the novelty of this work in the introduction of the paper.
Line 81-98
Point 2: Image or images of the obtained material should be added to the work to present the quality of CNTs obtained during other conditions of the synthesis in order to compare and to show the differences.
Response 2: We have added the FESEM images for bare monolith to show the original morphology of the support before coating with CNTs. The image was added in Figure 3.
Line 155-160
Point 3: Raman results in relation to the purity of the obtained nanotubes would enrich the work.
Response 3: We have included the RAMAN spectra to evaluate the purity of the obtained carbon nanotubes. The degree of disorder CNTs monolith was calculated by the intensity ratio of D band to the G band (ID/IG).
Line 285-298
Point 4: More discussion with the literature data can affect the improvement of scientific work - in particular as regards the characteristics of carbon nanotubes.
Response 4: We have added some more explanation regarding the XRD pattern and RAMAN curves.
Line 265-272, 285-296